# Hemin Promotes Higher Effectiveness of Aminolevulinic-Photodynamic Therapy (ALA-PDT) in A549 Lung Cancer Cell Line by Interrupting ABCG2 Expression

**DOI:** 10.3390/medsci12040066

**Published:** 2024-11-17

**Authors:** Anantya Pustimbara, Rahma Wirdatul Umami, Nurul Muhammad Prakoso, Anna Rozaliyani, Jamal Zaini, Astari Dwiranti, Shun-ichiro Ogura, Anom Bowolaksono

**Affiliations:** 1Cellular and Molecular Mechanisms in Biological System (CEMBIOS) Research Group, Department of Biology, Faculty of Mathematics and Natural Sciences, Universitas Indonesia, Depok Campus, Depok 16424, Indonesia; 2School of Life Science and Technology, Tokyo Institute of Technology, 4259 B47, Nagatsuta-cho, Midori-ku, Yokohama 226-8501, Japan; sogura@bio.titech.ac.jp; 3Department of Parasitology, Faculty of Medicine, Universitas Indonesia, Jakarta 10430, Indonesia; 4Department of Pulmonology and Respiratory Medicine, Faculty of Medicine, Universitas Indonesia, Persahabatan National Respiratory Referral Hospital, Jakarta 13230, Indonesia

**Keywords:** aminolevulinic acid, hemin, lung cancer, photodynamic therapy

## Abstract

Background/Objectives: Due to concerns about drug resistance and side effects, the discovery of improved drugs for lung cancer has attracted studies to find an effective and safe treatment. Aminolevulinic acid-mediated photodynamic therapy (ALA-PDT) is a cancer treatment with minimal side effects. However, ALA-PDT effectiveness can be hindered by ABCG2 and ABCB1 transporters impeding PpIX accumulation. Combining ALA with other substances can enhance PpIX accumulation. Hemin is a potential substance due to its antitumor properties and may be involved in regulating the ABCG2 and ABCB1 expressions. Methods: The objective of this report is to analyze the effects of administering a combination of hemin and ALA after 48 h on A549 lung cancer cells by quantifying cell viability, intracellular PpIX, and ROS accumulation, completed by ABCG2 and ABCB1 expressions. Results: Our data indicate that the combination of hemin and ALA followed by photoirradiation decreased the viability of A549 cells, which was due to increased intracellular PpIX and ROS. The expression of ABCG2 mRNA was significantly decreased after ALA-hemin treatment, while the ABCB1 mRNA expression increased. This result might suggest that ABCG2 plays a greater role than ABCB1 in regulating the PpIX accumulation in A549 lung cancer cells. Conclusions: The combination of ALA and hemin followed by photoirradiation offers a promising novel treatment for lung cancer, and further evaluations of this therapy are required.

## 1. Introduction

Lung cancer has been recognized as one of the most common types of cancer in many countries and is known to be one of the major leading causes of cancer mortality [1,2,3]. In Indonesia alone, the increasing number of lung cancer cases each year has led to an inevitable blow to the healthcare system, as the financial burden to manage the treatment of this group of cancer was reported to inflate to nearly USD 486 million in 2016 [4]. Although positive outcomes of the administration of small-molecule inhibitors to induce cytotoxicity and prevent the proliferation of non-small cell lung cancer (NSCLC) have been reported in clinical studies [5], the emergence of concerns regarding resistance and serious complications of these agents has promoted a significant interest in discovering an improved mode of treatment with higher effectiveness and relatively lower adverse events [6].

Aminolevulinic acid-mediated photodynamic therapy (ALA-PDT) is one of the most effective therapeutic methods due to its specificity and low side effects [7]. This method involves the administration of ALA as a prodrug to trigger a selective accumulation of protoporphyrin IX (PpIX) in cancer cells [8]. PpIX exhibits photosensitizing effects, allowing it to emit light when irradiated with specific wavelengths of light. In the presence of oxygen, this light can also induce the production of reactive oxygen species (ROS), which promotes cancer cell toxicity [9,10]. An issue with ALA-PDT is the inhibition of PpIX accumulation, the key factor of ALA-PDT, due to the activity of transporters in the cell. ABCG2 and ABCB1 are two multidrug transporters that are known to be the cause of this problem [11]. These transporters efflux PpIX out of the cell, reducing the effectiveness of ALA-PDT. Previous research targeted these transporters to enhance the effectiveness of ALA-PDT by combining other compounds to suppress the expression of these transporters in the cell [12].

Hemin is a porphyrin compound containing iron known as an inducer of HO1 (heme oxygenase 1). In our previous report, the combination of hemin in ALA-PDT has promoted higher effectiveness in inducing cellular toxicity in TMK-1 gastric carcinoma. The combined ALA-hemin followed by irradiation with a specific wavelength of light reduces the cell viability by altering iron homeostasis in TMK-1, leading to ROS-mediated cell death in this cell [13]. On the other hand, studies have reported that the accumulation of PpIX in promoting cellular toxicity in ALA-PDT is related to the regulation and activity of ABCG2 and ABCB1. The administration of specific inhibitors for ABCG2 and ABCB1 could result in decreased expression of these transporters and lead to the accumulation of PpIX inside the cell [14]. Previous studies have reported the involvement of ABCG2 inhibitors, Ko143 and FTC, to increase the effectiveness of ALA-PDT in breast and prostate cancer cell lines [15,16]. However, the application of combined ALA-hemin followed by photoirradiation for lung cancer and its molecular mechanism is still limited. In this study, we evaluated the potential of the ALA-hemin combination in enhancing the therapeutic effects of ALA-PDT against human lung cancer by regulating ABCG2 and ABCB1 expressions with hemin.

## 2. Materials and Methods

### 2.1. Cell Culture

The A549 lung carcinoma cell line was purchased from Riken BRC Cell Bank (Tsukuba, Ibaraki, Japan). The cells were cultured in RPMI 1640 supplemented with 10% fetal bovine serum (FBS) and 10% antibiotic–antimycotic solution (Nacalai Tesque, Tokyo, Japan) to support cell proliferation. The culture was maintained in a humidified incubator at 37 °C and 5% CO_2_. The culture management was performed by passaging when the cells reached 80% confluence.

### 2.2. Biochemicals

The 5-aminolevulinic acid (5-ALA) was obtained from Cosmo Bio Co. Ltd. (Tokyo, Japan). Hemin was purchased from Nacalai Tesque (Kyoto, Japan), while protoporphyrin IX dihydrochloride was purchased from Frontier Scientific, Inc. (Newark, DE, USA) as a standard substance for PpIX.

### 2.3. Cytotoxicity Assay

The A549 lung carcinoma cell line was seeded in a 96-well plate at a density of 5 × 103 cells/well and left overnight to allow proper adhesion. After incubating for 48 h with serial dilutions of ALA (4 mM) or hemin (80 μM), or the combination of ALA and hemin, the culture media were replaced, and the cells were irradiated with LED light (630 nm; 3.6 mW/cm^2^) for 5 min and left for further 24 h. The viability was assessed by adding 10 μL of MTT reagent to fresh culture media and then incubated for 4 h to allow the formation of formazan crystal, followed by lysing the cells with 10% SDS. After being left overnight, the cell viability was determined by measuring the absorbance at 570 nm and 650 nm using a Multiskan FC Microplate Photometer (Thermo Fisher Scientific, San Jose, CA, USA).

### 2.4. Hemin and ALA Treatment

To test the potential of inducing PpIX-mediated cytotoxicity in A549 cell lines, the research was conducted by incubating the cells either with ALA or hemin, or the combination of ALA and hemin. The ALA treatment was performed by incubating the cells with 1 mM ALA for 48 h. Another group of cells was incubated only with 40 μM hemin with the same duration. The combined administration of 1 mM ALA and 40 µM hemin was carried out during the 48 h of incubation, then followed by irradiation with LED light (630 nm; 3.6 mW/cm^2^) for 5 min. This treatment was conducted for subsequent analyses to compare the potential of the combined ALA and hemin to eliminate the A549 lung cancer cells in vitro.

### 2.5. High-Performance Liquid Chromatography (HPLC) to Measure PpIX Accumulation

The accumulation of PpIX in cells treated with ALA or hemin, or the combination of ALA and hemin, was conducted using the HPLC Prominence-I System (Shimadzu, Kyoto, Japan). A549 cells were seeded in a 6-well plate at a concentration of 0.3 × 106 cells per well, and then the treatment was performed accordingly. The cultures were washed with PBS prior to preparing the lysates using 0.1 M NaOH. The intracellular PpIX was extracted by running the lysates in a protein denaturant containing mobile phase A (1 M NH_4_CH_3_CO_2_ solution with 12.5% CH_3_CN at pH 5.2) and mobile phase B (50 mM NH_4_CH_3_CO_2_ solution with 80% CH_3_CN) at 1:9 (*v*/*v*) ratio. Centrifugation was performed twice at 10,000× *g* for 10 min at 4 °C to collect the supernatant, followed by the quantification of total protein concentrations with the Bradford assay (Bio-Rad Laboratories Inc., Hercules, CA, USA). The intracellular PpIX levels were quantified at 404 nm and normalized against the extracted total proteins.

### 2.6. Fluorescence Analysis of ROS and Intracellular PpIX

The intracellular accumulation of ROS and PpIX was conducted by fluorescence imaging with confocal laser microscopy. A549 cells were prepared in a 24-well plate and seeded at a density of 0.5 × 105 cells/well, then left overnight. After treatment with ALA or hemin, or the combination of ALA and hemin, the cells were washed with Hank’s balanced salt solution (HBSS) (-). To detect the accumulation of intracellular ROS, the cells were stained with 10 μM 2′,7′-dichlorodihydrofluorescein diacetate (DCFH-DA) and incubated for 30 min. The fluorescence signals of ROS were captured with Carl Zeiss LSM780 (Jena, Germany) at excitation of 488 nm and emission of 570–620 nm, while the intracellular deposition of PpIX was quantified at excitation of 405 nm and emission of 630–700 nm. The fluorescence images were taken with a 20 × magnification level, and the fluorescence data were analyzed by the ZEN software package of Zeiss (ver. 3.4.91.00000).

### 2.7. Analysis of Gene Expression

The total RNA of cells treated with ALA or hemin, or the combination of ALA and hemin, was extracted by using the NucleoSpin RNA II (Macherey-Nagel, Mannheim, Germany) according to the manufacturer’s instructions. The complementary DNA (cDNA) was synthesized using the PrimeScript RT Reagent Kit (TaKaRa Bio, Shiga, Japan). The expression of target genes listed in Table 1 was quantified with RT-qPCR using the SYBR Premix Ex Taq II (TaKaRa Bio, Shiga, Japan) on Thermal Cycler Dice Real Time System Single (version 6.1.). The relative mRNA expression levels of ABCG2 and ABCB1 were normalized against the expression of actin.

### 2.8. Western Blotting

The protein extraction was conducted on cells treated with ALA or hemin, or the combination of ALA and hemin, by adding lysis buffer A [50 mM Tris-HCl pH 7.4, 1% *v*/*v* Triton X-100, protease inhibitor cocktail (Nacalai Tesque, Tokyo, Japan)] and then homogenized with a 27-G needle. The homogenate was centrifuged at 1000× *g* for 10 min at 4 °C, and then the resulting supernatant was collected for a Bradford assay (Bio-Rad Laboratories, Richmond, CA, USA) to determine the protein concentration. The proteins were separated in 9.36% polyacrylamide gel electrophoresis and were electroblotted onto the Immobilon-P membrane (Millipore, Bedford, MA, USA). The membrane was incubated in 5% *w*/*v* skim milk in TBST [20 mM Tris-HCl pH 7.4, 150 mM NaCl, 0.05% *v*/*v* Tween20] at 4 °C for 1 h. The target proteins were captured using primary antibodies against human ABCG2 and ABCB1 (Santa Cruz Biotechnology; 1:200 dilution) and human actin (MP Biochemicals, CA, USA; 1:500 dilution). The secondary antibodies were targeted against the horseradish peroxidase (HRP)–conjugated anti-mouse IgG (Cell Signaling Technology, Danvers, MA, USA) and anti-rabbit IgG (Santa Cruz Biotechnology Ins., Dallas, TX, USA) at 1:3000 dilution. The expressions of ABCG2 and ABCB1 were quantified using the Western LightningTM Chemiluminescent Reagent Plus and Ultra (PerkinElmer Life and Analytical Sciences Ins., Waltham, MA, USA), then visualized in Amersham ImageQuant 800 (Cytiva Life Sciences, Shrewsbury, MA, USA).

### 2.9. Statistical Analysis

Data presented in this study are the mean values of each condition, with ±SD represented by error bars. Each dataset was subjected to one-way ANOVA and Tukey’s test to detect statistical differences in mean values between the treated and non-treated samples at four levels of significance. Data analysis and visualization were performed in GraphPad Prism 10.

## 3. Results

### 3.1. Reduced Viability of A549 After Combined Treatment of ALA and Hemin

The viability of A549 was performed using an MTT assay after being treated with ALA or hemin, or the combination of ALA and hemin, then followed by photoirradiation. Our results show that treatment with ALA reduced the viability to approximately 70%, while the combined treatment of 1 mM ALA and different concentrations of hemin resulted in the decline of viability to less than 5%. These data demonstrate a higher potential of the combined ALA and hemin in inducing cytotoxicity on A549 by more than 50% compared to treatment with ALA alone (Figure 1).

### 3.2. Elevated PpIX Accumulation in A549 After Co-Treatment of ALA and Hemin

To determine whether the significant reduction in cell viability was mediated by elevated accumulation of PpIX, its intracellular concentration was quantified by HPLC analysis. Our data illustrate that individual treatment with 1 mM ALA or 40 µM hemin indeed elevated the intracellular PpIX concentrations, although they were not statistically significant. Notably, the combination of 40 µM hemin and 1 mM ALA resulted in a significant increase in PpIX accumulation compared to control. This combination also produced higher intracellular PpIX than individual treatment with ALA or hemin (Figure 2). These findings bolster the proposition that a reduction in A549 viability may arise from the elevated PpIX accumulation in this cell, thereby promoting the activation of the cell death mechanism.

### 3.3. Combined ALA and Hemin Induces Cell Death via Intracellular ROS and PpIX Accumulation

The production of intracellular ROS in A549 exposed to the combination of ALA and hemin was monitored by live cell staining with 2′,7′-dichlorofluorescin diacetate (DCFH-DA) via confocal laser microscopy. In contrast to the control group, visual observation of fluorescence imaging demonstrated a higher intensity of 2′,7′-dichlorofluorescin in cells treated with the combination of ALA and hemin followed by PDT. A similar finding was also observed in regard to intracellular PpIX level, in which we observed a noticeable difference in the fluorescence intensity of PpIX in cells treated with both ALA and hemin. These findings were confirmed by quantitative analysis, which showed that the elevation of ROS and PpIX levels was statistically significant compared to control, indicating the higher cellular death event in this group may be mediated by the increased accumulation of intracellular ROS and PpIX in A549 cells treated with the combination of ALA and hemin (Figure 3).

### 3.4. ALA and Hemin Influence the mRNA Expression of ABCG2 and ABCB1

Gene expression analysis of ABCG2 and ABCB1 was conducted to validate that the intracellular PpIX accumulation was affected by the expression of these genes. The graphical representation in Figure 4 depicts a significant decrease in ABCG2 mRNA expression after hemin and the combined ALA-hemin addition, in which the latter group showed the lowest expression level. However, we observed a significant elevation of ABCB1 mRNA expression after being treated with hemin and the combined ALA-hemin, although this treatment was lower than hemin only. The significant reduction in ABCG2 mRNA expression might have a more dominant effect in regulating the accumulation of intracellular PpIX in cells treated with the combination of ALA-hemin; thereby, we argue that the involvement of hemin in the accumulation of PpIX was only attributed to the repression of ABCG2 rather than ABCB1.

### 3.5. Changes in ABCG2 and ABCB1 Protein Expression by Co-Treatment of ALA and Hemin

Western blot analysis was performed to further determine the protein expression of key transporters, ABCG2 and ABCB1, and to confirm our gene expression analysis. Our data demonstrate that the expression of ABCG2 was only reduced following ALA-hemin treatment, despite not being statistically significant. These data also show an increased expression of ABCB1 after being treated with ALA or hemin, and the combination of ALA and hemin, suggesting that the action of hemin to induce PpIX accumulation was mediated by repressing the ABCG2 transporter rather than both of these transporters, which are also consistent with our RT-qPCR results (Figure 5).

## 4. Discussion

Aminolevulinic acid, or ALA, is produced naturally in cells as a key precursor in the heme-biosynthesis pathway [17]. ALA undergoes several stages of conversion to other molecules in this pathway to become PpIX, which has photosensitization and fluorescence activity [18]. The basic concept of ALA-PDT is the administration of exogenous ALA that will be taken up by transporters into the cell and then induce selective accumulation of PpIX in cancer cells due to different metabolisms between cancer and normal cells [19]. The cell death is then induced by photoirradiation to produce reactive oxygen species, leading to apoptosis [20,21].

Despite its advantages of minimum side effects and high specificity, ALA-PDT’s effectiveness in killing cancer cells can be reduced by transporter activity that effluxes PpIX out of the cell [22]. ABCG2 is commonly known as the PpIX efflux transporter, and its expression affects the PpIX accumulation [15,23,24]. The previous study also revealed another transporter, ABCB1, that plays an important role in PpIX efflux [25]. Previous research aimed to downregulate or inhibit the activity and expression of these transporters to escalate the PpIX accumulation by combining ALA with other substances and targeting some pathways that are related to these transporters [25,26,27]. We analyzed the potential involvement of hemin combination to improve the efficacy of ALA-PDT to induce cell death in A549 lung cancer and hypothesized that it regulates the PpIX accumulation by repressing the ABCG2 and ABCB1 expression.

Our viability assay indicates the co-administration of hemin with ALA significantly improves the cellular toxicity in A549 lung cancer cells compared to the individual treatment with ALA. Previous research has reported that hemin itself exhibits inherent capabilities to impair cancer cell viability and impede tumor progression, migration, and invasion, while also deterring metastasis [28]. In another research study, hemin has been implicated in triggering ferroptosis, a form of programmed cell death distinct from apoptosis, primarily facilitated by iron-dependent oxidative processes [29]. Hemin-induced ferroptosis in A549 cells entails the accumulation of reactive oxygen species (ROS) and lipid peroxidation, leading to cellular demise. This cascade is orchestrated by hemin’s proteasomal activity, facilitating ferritin degradation and consequent iron mobilization, thereby amplifying ROS production and lipid peroxidation, which may induce specific cellular toxicity in cancer cells [30].

To determine whether the cellular toxicity was induced by the increased accumulation of PpIX, its intracellular concentration was quantified by HPLC analysis. Our result shows that co-treatment with ALA and hemin substantiates the notion that the reduction in cell viability may stem from factors augmenting intracellular PpIX accumulation. This argument was in accordance with our fluorescence imaging, in which a significant elevation in the production of intracellular ROS and PpIX was observed in cells treated with the combination of ALA and hemin. This result provides the mechanistic explanation of ROS-induced cellular toxicity in A549 cancer cells, which was mediated by the intracellular accumulation of PpIX [31]. It has been known that PpIX, upon light absorption, undergoes excitation and subsequently generates cytotoxic ROS, inducing mitochondrial damage and tumor cell apoptosis, ultimately culminating in cell death [21].

Several factors, including the transporter activity of PpIX, may contribute to the level of PpIX accumulation inside the cells [32]. Gene expression analysis and Western blotting were performed to observe the roles of the two transporters, ABCG2 and ABCB1. In this study, we observed noticeable reductions in the mRNA and protein expression of ABCG2 in cells following the addition of the combined ALA and hemin, indicating that hemin might play a key role in repressing the expression of ABCG2 to prevent the efflux of intracellular PpIX, thereby participating to effectively increase the PpIX-mediated cellular toxicity in A549 lung cancer cells [33]. In contrast to ABCG2, we noted elevations of ABCB1 expressions after being treated with hemin and the combination of ALA and hemin, which would suggest that the ABCB1 might have a lesser impact in regulating the concentration of PpIX in our study. This might be explained by a significant increase in intracellular ROS and PpIX levels, which was still observed during the escalation of ABCB1 expression in this group.

There may be other mechanisms regarding increased expression of ABCB1 after the combined treatment, such as hemin might induce genetic and epigenetic alterations in cells that contribute to the increase in ABCB1 expression [34]. Alternatively, hemin might serve as a substrate of ABCB1 as doxorubicin, given its role in the effluxion of various cancer substances and drugs [35]. However, transcriptome-level studies could be more informative to provide the global expression pattern regarding the molecular mechanism of ALA-hemin in promoting cell death in the A549 cell line. Despite further investigations in animal and human subjects being required, experimental data regarding the effectiveness of ALA-hemin in this study offer a promising candidate to provide more treatment strategies for patients with lung cancer.

Nevertheless, this study has shown that the effectiveness of ALA-PDT in inducing cellular toxicity in A549 lung cancer cells was increased by combining ALA-hemin to accelerate the accumulation of PpIX, which would induce ROS-mediated cell death. We argue that hemin altered the expression of ABCG2 to prevent the efflux of intracellular PpIX to maintain its toxic level inside the cells. We also speculate that although its expression was increased, ABCB1 made little contribution in reducing the level of PpIX in A549 cells after ALA-hemin treatment; this was explained by the persistent accumulation of ROS and intracellular PpIX in our study. The contribution of ABCB1 in transporting out the intracellular PpIX might also be influenced by cell type, as this phenomenon was also reported by another study, in which the inhibition of ABCB1 did not participate in the accumulation of PpIX in MDA-MB-231 cells [36].

## 5. Conclusions

To summarize, our results may also indicate that ABCG2 plays a greater role than ABCB1 in PpIX accumulation, as increased PpIX concentration in our data might be explained by repression of ABCG2, both in mRNA and protein level, after the combination of ALA and hemin treatment. This proposition should be investigated in our future research by conducting an ABCG2 knockout study. The involvement of other genes that have the role of regulating the other heme-synthesis pathway has to be investigated. Moreover, pathway analysis of cellular death mechanisms will be considered to obtain a complete understanding of potential targeted cell death pathways to selectively induce toxicity in cancer cells.

## Figures and Tables

**Figure 1 medsci-12-00066-f001:**
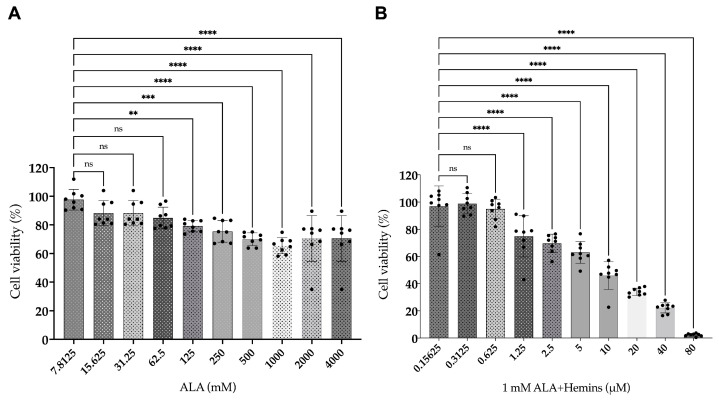
Determination of A549 viability using MTT assay after (**A**) ALA and combined (**B**) ALA-hemin addition followed by light irradiation. The most toxic concentration of ALA reduced the cell viability to ~70%, while the highest concentration of the combined treatment reduced the viability to 2.2%. (*n* = 8, ns, not significant; **, *p* < 0.01; ***, *p* < 0.001; and ****, *p* < 0.0001). The distribution of each experimental replication is presented as the black dots.

**Figure 2 medsci-12-00066-f002:**
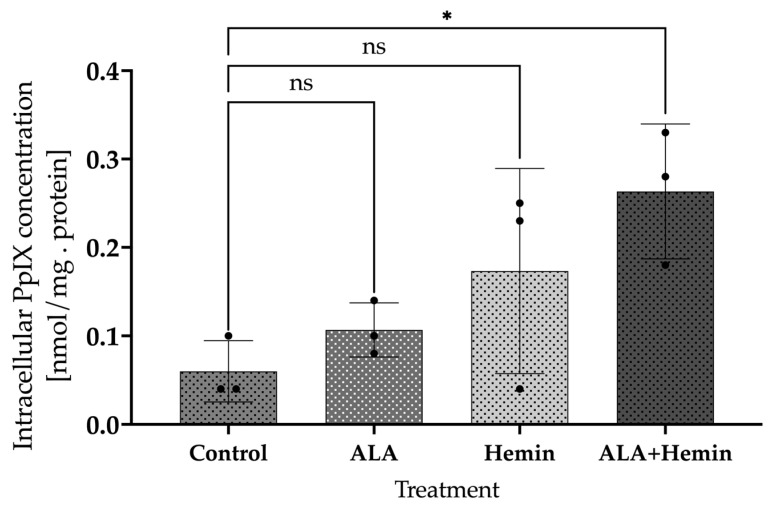
Changes in intracellular PpIX accumulation after ALA and hemin addition. A one-way analysis of variance (ANOVA) and Tukey’s test were performed to detect differences in mean values between treated and non-treated samples (*n* = 3, ns, not significant; *, *p* < 0.1). The distribution of each experimental replication is presented as the black dots.

**Figure 3 medsci-12-00066-f003:**
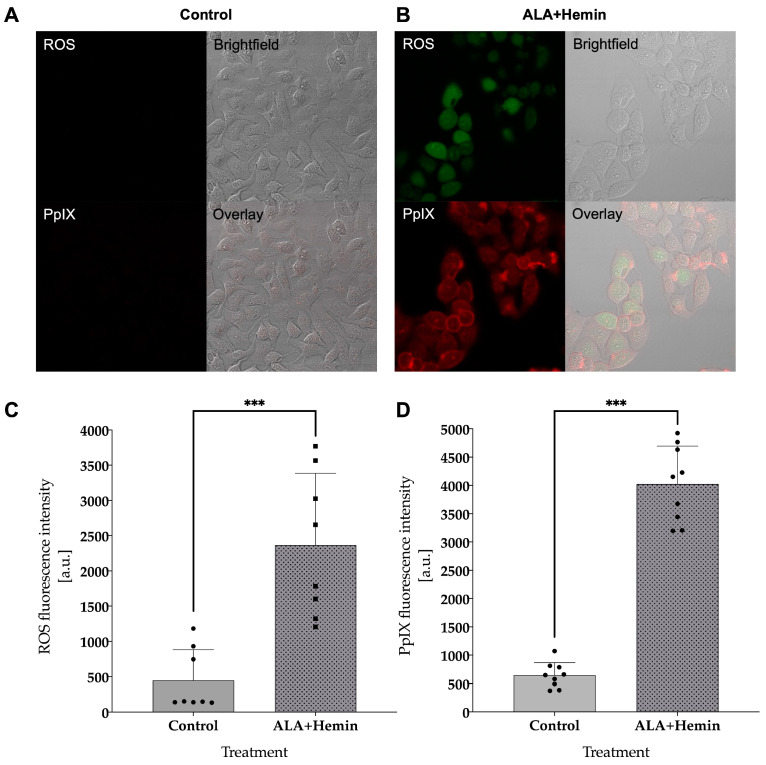
Confocal microscope observation of PpIX and ROS in (**A**) control A549 cells and (**B**) after ALA-hemin treatment. The quantitative measurement of PpIX and ROS in (**C**) the control group and (**D**) after being treated with ALA-hemin. Quantitative data were expressed as mean with ±SD, analyzed with one-way analysis of variance (ANOVA), and Tukey’s test to confirm differences in the statistical significance of each treatment (*n* = 8, ***, *p* < 0.001). The distribution of each experimental replication is presented as the black dots.

**Figure 4 medsci-12-00066-f004:**
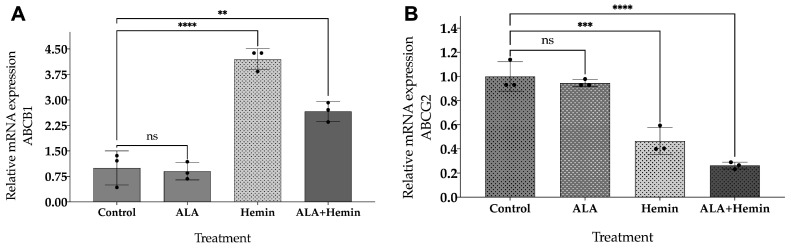
Changes in relative gene expression of two PpIX transporters, (**A**) ABCB1 and (**B**) ABCG2, after ALA and hemin addition. A one-way analysis of variance (ANOVA) and Tukey’s test were performed to detect differences in mean values between the treated and non-treated samples (*n* = 3, ns, not significant; **, *p* < 0.01; ***, *p* < 0.001 and ****, *p* < 0.0001). The distribution of each experimental replication is presented as the black dots.

**Figure 5 medsci-12-00066-f005:**
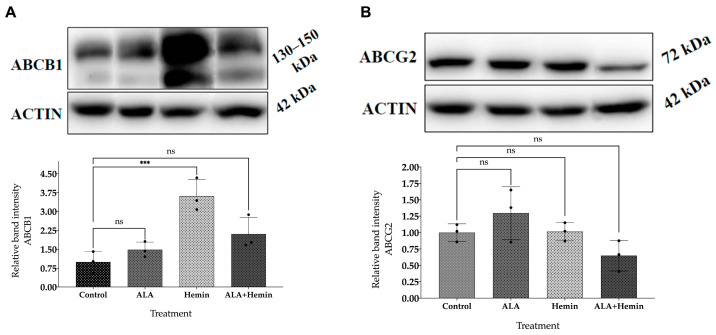
Differences in (**A**) ABCB1 and (**B**) ABCG2 protein expression following ALA and hemin addition. The Western blotting images shown in this figure were derived from the complete image provided in Appendix A. A one-way analysis of variance (ANOVA) and Tukey’s test were performed to detect differences in mean values between treated and non-treated samples at two levels of significance (*n* = 3; ns, not significant; ***, *p* < 0.001). The distribution of each experimental replication is presented as the black dots.

**Table 1 medsci-12-00066-t001:** List of specific primers for RT-qPCR analysis.

Gene	Forward Primer	Reverse Primer
*ABCG2*	5′ GCAACCATCAATTCAGGTCAAGA ‘3	5′ GAAACACAACACTTGGCTGTAGCA ‘3
*ABCB1*	5′ GGAAGCCAATGCCTATGACTTTA’3	5′ GATGACGTCAGCATTACGAACTGTA ‘3
*ACTIN*	5′ TGGCACCCAGCACAATGAA ‘3	5′ CTAAGTCATAGTCCGCCTAGAAGCA ‘3

## Data Availability

Data are contained within the article and Appendix A.

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
