# Peer review of "Hemin Promotes Higher Effectiveness of Aminolevulinic-Photodynamic Therapy (ALA-PDT) in A549 Lung Cancer Cell Line by Interrupting ABCG2 Expression"

_medsci, 2024, doi:10.3390/medsci12040066_

Round 1
Reviewer 1 Report
Comments and Suggestions for Authors
This manuscript investigates the authors' hypothesis that hemin could enhance the effectiveness of aminolevulinic-photodynamic therapy (ALA-PDT). Indeed, the authors demonstrate that, in A549 lung cancer cells, hemin significantly decreases the viability of cells upon ALA-PDT treatment. Furthermore, the authors demonstrate the accumulation of both reactive oxygen species and PPIX in cells upon ALA-PDT treatment upon addition of hemin, which the authors link to lower mRNA levels of the PPIX transporter ABCG2. The authors present a well-written manuscript that should be of interest to those interested in ALA-PDT treatment for cancer as well as those interested in heme homeostasis.
A major critique of the manuscript is that the authors in both figure 2 and figure 5 draw conclusions based on data that they state is different than the control "despite being not statistically significant". Indeed, the data presented in figure 5 indicates that ABCG2 protein levels are NOT significantly different with hemin treatment, yet the authors report in the abstract and the text that ABCG2 protein levels are altered upon the addition of heme. The assumption appears to be that the mean levels are different, but this conclusion cannot be drawn from the data because of the statistics.
Related to the above comment, the authors do not report the number of times any of the experiments have been replicated (n). In addition to reporting the n value, the authors should plot the individual data points on all of the bar graphs throughout the manuscript.
In figure 1, a control in which heme alone (without ALA) would have provided a lot of consistency with the rest of the manuscript.
In figure 3, the figure legend is not consistent with the data presented. The bar graphs below should be labeled as C and D since they are comparing data in parts A and B.
In figure 5, the western blots appear to be cropped from the supplementary figure. If this is the case, this information should be indicated in the figure legend for transparency.
Reviewer 2 Report
Comments and Suggestions for Authors
The manuscript aims to presents the results of experiments aimed at elucidating the role of hemin in photodynamic efficiency of aminolevulonic acid in lung cancer cells in vitro. The manuscript presents the data concerning ALA mediated decrease in viability of A549 cells, as well quantitative determination of protoporphyrin IX amount retained within cells following treatment with ALA and hemin. Moreover authors described changes in gene and protein expression of two transporters responsible for ALA efflux out of cells exposed to ALA and ALA with hemin.
The main strength of the manuscript are as follows: clearly presented research hypothesis and well explained research gap, an appropriate selection of methods, proper statistical analysis. Data presentation is clear and consistent, the article is well written, the authors clearly discussed their findings with previously performed studies.
I would raise some points that should be better addressed by authors:
1) The idea that ALA efficiency could be enhance by reducing efflux has been discussed before. There has been some previous work on ABCG2 inhibitors and their influence on Protoporphyrin IX accumulation in cancer cells, e.g. lung cancer. Although a small number of such molecules has been identified so far, the observed phenomena are in-line with authors’ findings. These papers could be cited in Introduction in order to draw a bigger picture of this research area: https://doi.org/10.1158/0008-5472.CAN-15-1484, https://doi.org/10.1038/srep13298, 10.1016/j.pdpdt.2021.102452
2) The authors point out some alternative mechanisms that could still be responsible for the improvement of photodynamic activity of ALA in the presence of hemin in line 317. Besides that are there some more implications for future research that authors could draw from the present study?
3) ALA and hemin are both registered drugs, although hemin is used in porphyria. Could the present research have practical implications in new cancer treatment modalities, given the results of beneficial effect of their co-administration in cancer cells?
I would also highlight some minor points:
1) The word ‘research’ is singular and thus the sentence in line 57 should be corrected
2) The word ‘either’ should be used instead of ‘whether’ in line 102
3) Abbreviation ‘HBSS’ was not explained in line 128
